# Cryo-EM observation of the amyloid key structure of polymorphic TDP-43 amyloid fibrils

Kartikay Sharma [1] ✉, Fabian Stockert [1], Jayakrishna Shenoy [2], Mélanie Berbon[2], Muhammed Bilal Abdul-Shukkoor[2], Birgit Habenstein [2], Antoine Loquet [2], Matthias Schmidt [1] & Marcus Fändrich [1]

The transactive response DNA-binding protein-43 (TDP-43) is a multi-facet protein involved in phase separation, RNA-binding, and alternative splicing. In the context of neurodegenerative diseases, abnormal aggregation of TDP-43 has been linked to amyotrophic lateral sclerosis and frontotemporal lobar degeneration through the aggregation of its C-terminal domain. Here, we report a cryo-electron microscopy (cryo-EM)-based structural characterization of TDP-43 fibrils obtained from the full-length protein. We find that the fibrils are polymorphic and contain three different amyloid structures. The structures differ in the number and relative orientation of the protofilaments, although they share a similar fold containing an amyloid key motif. The observed fibril structures differ from previously described conformations of TDP-43 fibrils and help to better understand the structural landscape of the amyloid fibril structures derived from this protein.

Amyloid fibrils are fibrillar polypeptide aggregates with a cross-β sheet structure[1,2]. These sheets are defined by an orientation of the β-strands that is roughly perpendicular to the fibril main axis[3], such that multiple protein molecules can be stacked up via backbone hydrogen bonds in the direction of the main fibril axis[4,5]. The relative orientation of two hydrogen-bonded β-strands in cross-β sheets is parallel in available cryo-EM structures[6]. The strands of the fibril protein are connected within each molecular layer of the fibril via arcs, similar to β-solenoid structures[7]. In several cases, there was evidence for a fibril protein fold that remotely resembles the Greek key structure in globular proteins[8,9]. This motif of amyloid fibrils is termed an amyloid key[10]. An amyloid key represents an arrangement of β-strands but in contrast to a Greek key, the strands interact through their side chains and not through backbone hydrogen bonds. The fibril protein stacks define the fibril protofilaments (PFs), which represent filamentous substructures of amyloid fibrils.

Amyloid fibril samples are typically polymorphic and contain multiple fibril morphologies[11,12]. These fibril morphologies may differ in the number of PFs, in the orientation of the PFs relative to one another, and in the PF substructure, that is, in the fold of the fibril protein[12]. This spectrum of fibril morphologies present in a sample can be influenced by the incubation time[13] as well as by the physical and chemical conditions under which aggregation occurs[14]. Amyloid fibril polymorphism has been of scientific interest in the context of the formation of different disease variants, where it has possible ramifications for diagnosis and therapy[15]. Examples of different amyloid fibril morphologies in different disease variants are type A and B transthyretin fibrils in hereditary ATTR amyloidosis[16], different serum amyloid A fibrils in the glomerular and vascular forms of systemic AA amyloidosis[10,17], and different α-synuclein, tau and Aβ-derived fibril structures in different neurodegenerative conditions[18–21].

In this research, we have studied a sample of fibrils formed from the transactive response DNA-binding protein-43 (TDP-43). TDP-43 is a multifunctional protein that is involved in the transcription, processing, and translation of mRNA[22,23]. It consists of an N-terminal (NT) domain, a low-complexity (LC) C-terminal domain as well as a nuclear localization signal (NLS) sequence and two RNA recognition motifs (RRM1 and RRM2) (Fig. 1a). The LC domain is rich in Gly and Gln/Asn

[1]Institute of Protein Biochemistry, Ulm University, 89081 Ulm, Germany. [2]University of Bordeaux, CNRS, Bordeaux INP, CBMN, UMR 5248, IECB, Pessac, France. ✉e-mail: kartikay.sharma@uni-ulm.de

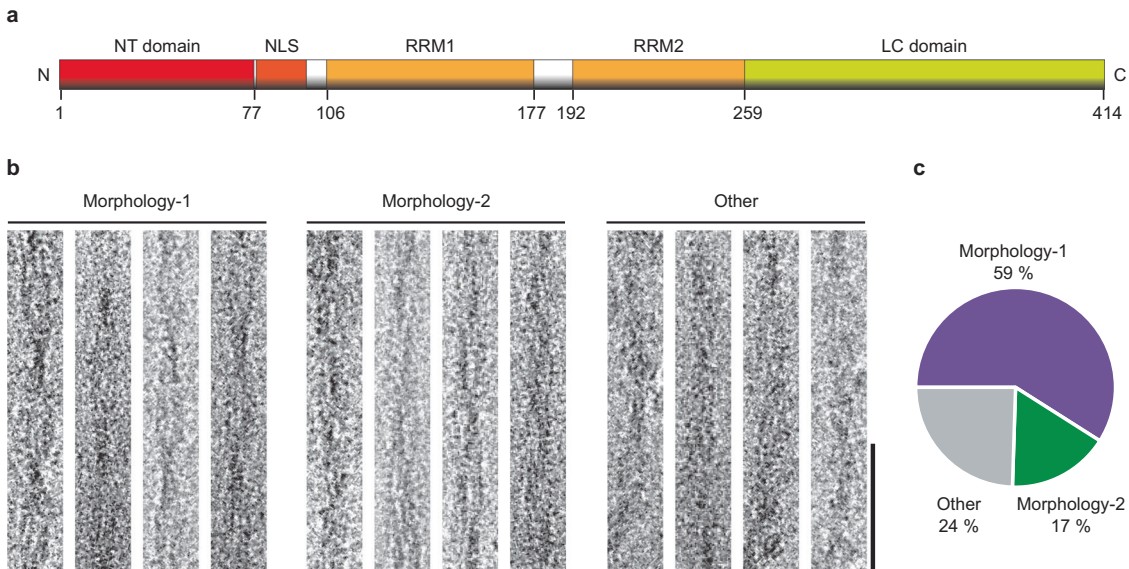

**Fig. 1 | Morphological composition of the analyzed sample. a** Schematic representation of the primary structure of TDP-43 protein. Boxes represent different motifs or domains of the protein (first and last residues are labeled). **b** Cryo-EM images of Morphologies-1 and 2, and of the other fibril morphologies in the sample. Scale bar: 50 nm. The images are representative of the fibrils from a total of 1641 micrographs. **c** Relative abundance of Morphologies-1, 2, and others in the cryo-EM data set ($n = 200$).

residues, intrinsically disordered and believed to drive the aggregation of TDP-43 protein[22,24,25]. Aggregated TDP-43 is associated with amyotrophic lateral sclerosis, frontotemporal lobar degeneration, as well as other diseases[26]. The spectrum of different TDP-43-associated pathologies is thought to be linked to multiple conformations or fibril polymorphs of TDP-43[27,28]. Most structural studies of TDP-43 amyloid fibrils have been carried out with fibrils that were obtained by in vitro incubation of truncated constructs of TDP-43 protein[29–32]. However, the rationale behind the observed fibril polymorphism originating from the full-length protein is still missing.

Using cryo-EM, we have investigated a sample of TDP-43-derived amyloid fibrils that were formed in vitro from full-length TDP-43. We find that the protein is able to form a polymorphic spectrum of fibril morphologies. The observed fibril structures are constructed from similarly folded fibril proteins, exhibiting an amyloid key motif. The observed fibril structures differ from previously described TDP-43 structures and provide a possible mechanistic explanation for the formation of this structural element in amyloid fibrils.

## Results

### TDP-43 forms polymorphic amyloid fibrils

Full-length TDP-43 (residues 1-414) was purified according to a previously described procedure[33] and incubated in 10 mM 4-(2-hydroxyethyl)−1-piperazineethanesulfonic acid (HEPES), pH 7.4, containing 40 mM NaCl for 3 days at room temperature under shaking. The purification and fibril formation procedure yielded full-length TDP-43 protein (Supplementary Fig. 1a, b), although it is possible that proteolytic truncation may occur subsequently. Transmission electron microscopy (TEM) shows that the fibrils possess a well-resolved crossover structure, indicating that the fibrils are twisted (Supplementary Fig. 1c). Platinum side shadowing additionally shows that the twist is left-handed (Supplementary Fig. 1d). Plunge freezing of the fibrils and imaging under cryogenic conditions reveals a heterogeneous sample composition, containing different fibril morphologies (Fig. 1b). This observation is in line with previous solid-state nuclear magnetic resonance studies performed on TDP-43-derived fibrils obtained under similar conditions[33].

Two relatively abundant fibril morphologies could be discerned in our samples, which we term here Morphology-1 and Morphology-2 (Fig. 1b). Morphology-1 has a width of $11.5 \pm 0.8$ nm and well-resolved cross-overs at a distance of $46.9 \pm 4.5$ nm ($n = 50$, Fig. 1b). Morphology-2 show a width of $10.4 \pm 0.6$ nm ($n = 50$, Fig. 1b), while the fibril cross-overs are hardly visible and no crossover distance could be measured. Approximately 59% of the fibrils visible in the sample correspond to Morphology-1 and 17% to Morphology-2 (Fig.1c). The remaining fibrils were morphologically heterogeneous and could not be analyzed in further detail.

### Cryo-EM structures of three TDP-43 fibril morphologies

Subjecting the cryo-EM images of Morphology-1 to a reconstruction of the three-dimensional (3D) map resulted in the subdivision of the extracted fibril segments during 3D classification. The two distinct 3D classes were identified and refined. We hereafter refer to these two structures as Morphologies-1a and 1b (Fig. 2 and Supplementary Fig. 2). The 3D maps reached spatial resolutions of 3.76 Å (Morphology-1a) and 4.05 Å (Morphology-1b), based on the 0.143 Fourier Shell Correlation (FSC) criterion (Fig. 2 and Supplementary Fig. 3b, c). In the case of Morphology-2, it was not possible to identify more than one 3D class, and the 3D map of this fibril structure was refined to a resolution of 3.86 Å (Fig. 2 and Supplementary Fig. 3b, c).

All reconstructions used a left-hand fibril twist (Supplementary Fig. 1d). The resulting fibrils are polar and differ in the number of the fibril protein stacks as well as in their symmetry. Morphology-1a is C2-symmetrical and consists of two structurally equal, in-register protein stacks (Fig. 2 and Supplementary Fig. 4a). Morphology-1b contains two protein stacks that are not equal (termed here P1 and P2) and staggered in the direction of the fibril axis. Moreover, the two stacks show a different orientation relative to one another (Fig. 2 and Supplementary Fig. 4a). Hence, this fibril structure is C1-symmetrical. In the case of both Morhphology-1a and 1b, extra densities were observed near the contact sites of two protofilaments (Supplementary Fig. 3b). At the current resolution of the 3D maps, we cannot comment on the origin of these extra densities. However, it can be possible that they arise due to terminal regions of the fibril protein. Morphology-2, finally, contains three structurally equal, in-register fibril protein stacks, and the fibril is

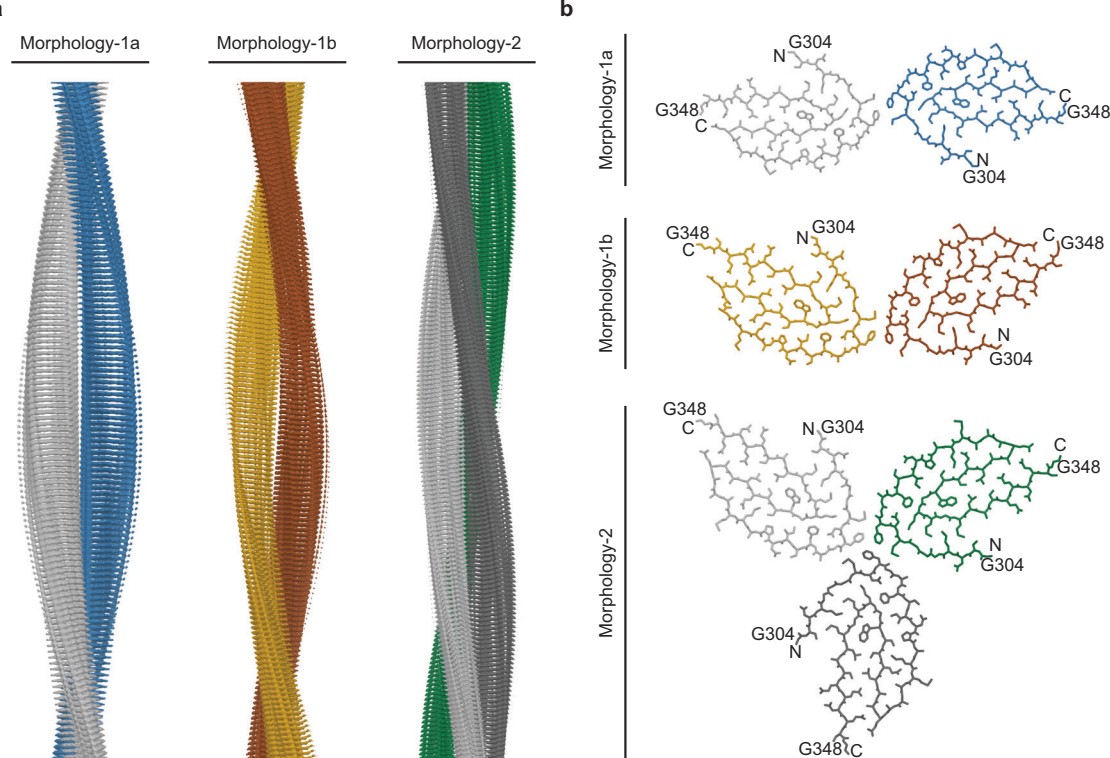

**Fig. 2 | Cryo-EM 3D maps and molecular models of TDP-43 amyloid fibrils. a** Side views of the 3D maps of Morphologies-1a, 1b and 2. **b** Cross-sectional views of the molecular models (stick representation). The first and the last amino acid residues of the fibril core are labeled. The fibril protein stacks are consistently colored coded in all panels. Morphology-1a: stack 1: light gray; stack 2: blue. Morphology-1b: stack 1: brown; stack 2: yellow. Morphology-2: stack 1: light gray; stack 2: dark gray; stack 3: green.

C3-symmetrical (Fig. 2 and Supplementary Fig. 4a). Despite these differences, we find the interfaces between the adjacent protein stacks to be formed by the same two amino acid residues (Met311 and Phe313) in the three fibrils (Supplementary Fig. 4b).

### Evidence for structural breaks in Morphology-1

In the next step, we tested whether Morphologies-1a and 1b represent truly different fibril morphologies or whether they may rather be classified as sub-morphologies that may coexist within mixed fibrils. The latter case was previously reported for a sample of light-chain derived amyloid fibrils from systemic AL amyloidosis[34]. Therefore, we analyzed the location of the fibril segments that led to the two 3D maps in our micrographs. We find that the majority (~66%) of the fibrils visible in our micrographs are mixed and contain segments from both structural forms (Fig. 3). Only ~34% of the fibrils are morphologically pure and contain only Morphology-1a segments or only Morphology-1b segments (Fig. 3). The abundance of the segments corresponding to the two structural forms (1a and 1b) varies almost continuously within mixed fibrils from one to zero (Fig. 3b). The segments of the two structural forms are not randomly scattered throughout the mixed fibrils but occur clustered into distinct $z$-axial regions of variable length (Fig. 3a). These features imply that Morphologies-1a and 1b do not represent systematically distinct fibril morphologies but sub-morphologies that are able to coexist in mixed fibrils.

### The fold of TDP-43 protein fibrils

The three fibril structures contain essentially the same fibril protein conformation that is formed by 45 residues (Gly304 to Gly348) from the LC domain (Fig. 4a). All peptide bonds are present as trans isomers. The fibril protein fold is all-beta, compact, and devoid of large internal cavities. The fibril proteins participate in the formation of

intermolecular cross-β sheets with parallel strand-strand interactions along the fibril axis (Fig. 2a). Our assignment of the β-strands is based on two criteria: all residues of a strand lie within the β-sheet region of the Ramachandran plot and all are involved in the formation of β-sheet backbone hydrogen bonds (O-H distance <3 Å). Although the protein fold is very similar in the three fibrils, there are differences in the exact geometry of the polypeptide backbone that translate into differences in the definition of the β-strands, according to our two criteria (Fig. 4b). These differences imply that the fibril protein conformation is able to adapt to its topological position within the overall fibril structure. The fibril core is devoid of buried ion pairs (Supplementary Fig. 5a). All fibril proteins contain arch that is formed by residues Ala328-Ala341, while residues Gly304-Ala326 are wrapped around this arch (Fig. 5a). Therefore, the protein fold shows an amyloid key motif, which is similar to a Greek key except that the polypeptide chains interact through their side chains rather than through backbone hydrogen bonds (Supplementary Fig. 5b).

### Location of the most aggregation-prone segments

To evaluate the formation of this structure, we used different computational programs to identify the most aggregation-prone regions within the sequence of TDP-43 protein. We find that the most aggregation-prone regions do not necessarily lie within the residues that form the fibril core but within the NT domain and two RRMs (Supplementary Fig. 6). As both the NT domain and the RRMs are known to be folded under near-physiological conditions[35–37], these TDP-43 segments are not available for the formation of an intermolecular β-sheet structure; that is, their intrinsic aggregation propensity is overruled by the efficient folding of these parts of the protein. In the LC domain, by contrast, which is known to be intrinsically disordered[22,24], the two most aggregation-prone regions (as defined by an aggregation score of three or more) occur at residues

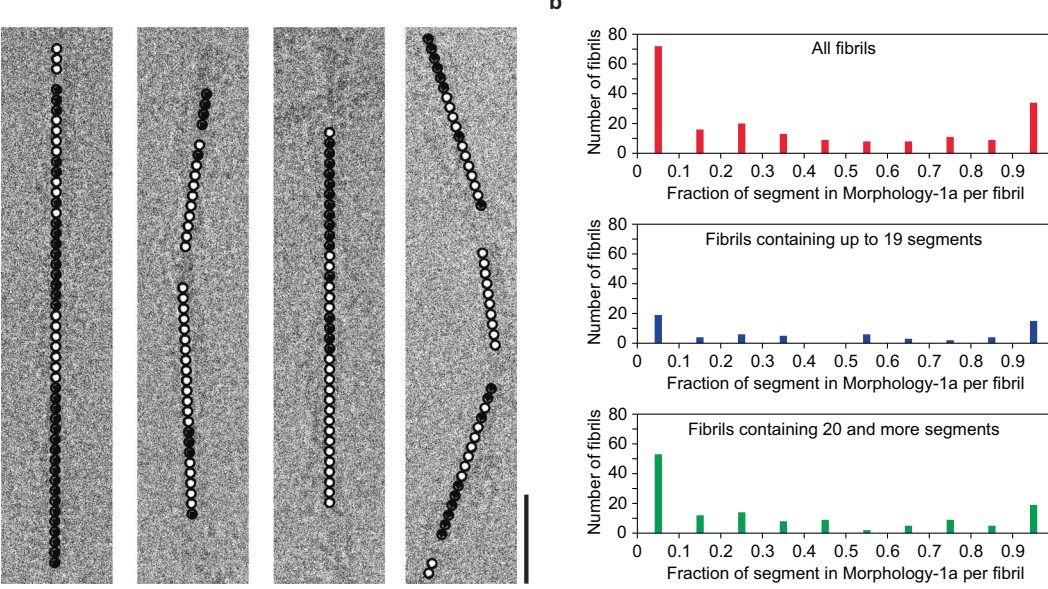

**Fig. 3 | Evidence for structural breaks in Morphology-1. a** Cropped cryo-EM images show the Morhphology-1a (black dot) and Morphology-1b (white dot) segments which occur clustered into distinct z-axial regions. Morhphology-1a and 1b segments were found to coexist in 132 out of 200 analyzed fibrils. Scale bar: 25 nm. **b** The fraction of Morphology-1a segments, per fibril (200 fibrils, top). Evaluation of a subset of fibrils with 1–19 segments (64 fibrils, middle) and of fibrils with 20 or more segments (136 fibrils, bottom).

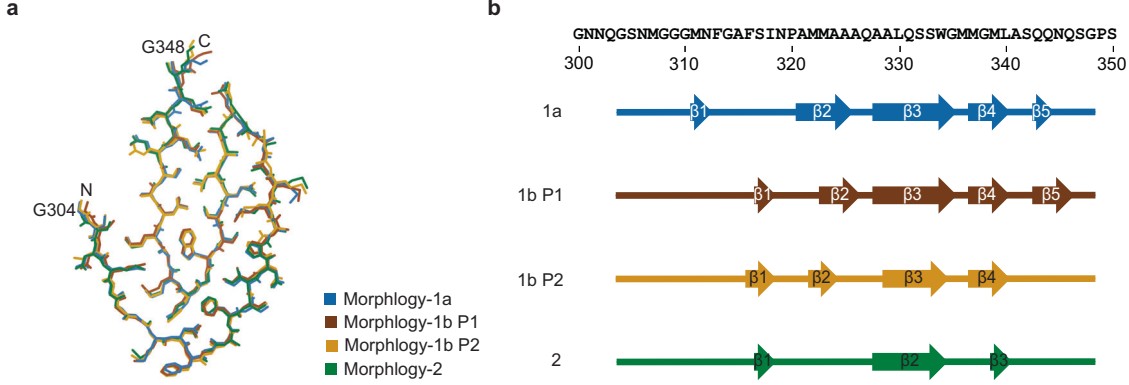

**Fig. 4 | Fibril protein fold in the three fibril morphologies. a** Super-imposition of the fibril proteins of Morphology-1a, Morphology-1b and Morphology-2 (stacks P1 and P2). The first and the last amino acid residues of the fibril core are labeled. Note that differences in the side chain geometries may have arisen from the modeling. **b** Sequence of residues Gly300 to Ser350 of TDP-43 drawn side-by-side with schematic representations of the fibril protein core in the different fibril morphologies as indicated in the panel. Arrows indicate β-strands. The color coding of the fibril morphologies is kept consistent in both panels.

Phe313 to Ile318 and Gly335 to Met339 (Supplementary Figs. 6 and 7a). Both regions are paired in our structure (Supplementary Fig. 7b), suggesting that their pairing might drive the aggregation of TDP-43 protein in our samples. Similar observations and conclusions were previously obtained for the aggregation-prone regions in immunoglobulin light chains and their effect on amyloid fibril formation[38].

## Discussion

In this study, we have investigated the structure of TDP-43-derived amyloid fibrils formed in vitro. We found three different fibril structures in our sample that share the same basic fibril protein fold and differ in the number and relative arrangement of the fibril protofilaments (Fig. 2). Morphologies-1a and 1b do not represent systematically distinct fibril structures and may coexist within mixed fibrils (Fig. 3). The fibril proteins show the same basic fold in all three fibril structures

and contain an amyloid key motif (Figs. 2 and 4). The amyloid key is defined by a topological trace of the polypeptide chain that roughly resembles the Greek key motif in globularly folded protein structures (Supplementary Fig. 5b).

The Greek key is defined as an element of β-sheet structure with a +3, −1, −1 or +1, +1, −3 topology, based on the Richardson nomenclature[39]. It contains four (or five) β-strands, which form—at least in principle—one common β-sheet; that is, the β-strands are connected via backbone hydrogen bonds (Supplementary Fig. 5b). These properties differ starkly from the situation in the amyloid key, where the β-strands do not interact via backbone hydrogen bonds but rather via side chain–side chain interactions (Supplementary Fig. 5b). As a consequence, the strands of an amyloid key do not form a common β-sheet but each strand participates in the formation of a different β-sheet. The amyloid key motif has been redundantly described in

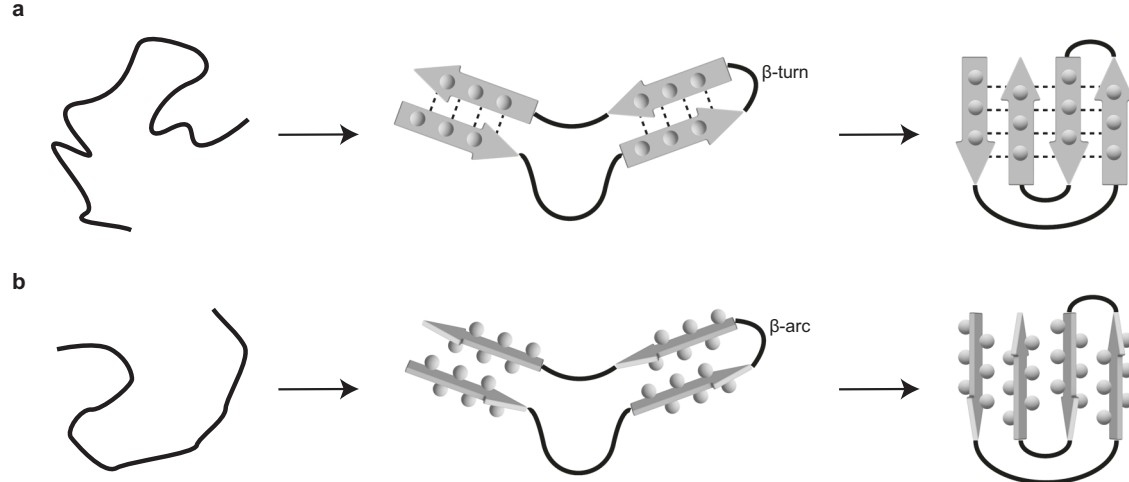

**Fig. 5 | Mechanism of the formation of the amyloid key and the Greek key.**
**a** Schematic representation of the previously proposed mechanism of the formation of a Greek key[59]. **b** Schematic representation of the possible mechanism for the formation of an amyloid key. Thick continuous lines: the polypeptide chain; arrows: β-strands; spheres: side chains; dotted lines: backbone hydrogen bonds.

the context of amyloid fibril structures[8,9], although its definition from the β-sheet structure is less clear and the motif is often confused with a Greek key.

None of the previously described TDP-43 conformations and fibril structures shows an amyloid key motif[29–31,40,41]. Some of these fibrils were formed from relatively short fragments of TDP-43 protein that lack Asn312 to Gln346, which form the core of our fibrils (Supplementary Fig. 8). In case of the in vitro formed fibrils, it is interesting that the majority of these fibrils show a pairing of the two most aggregation-prone regions of the LC domain and an arc or kink at residues Ala325 to Ala329 that allows the polypeptide chain to fold back onto itself such that the two aggregation-prone regions become paired (Supplementary Fig. 9). These structures contained a U-shaped conformation of this polypeptide segment, but only our structures contain an amyloid key (Supplementary Fig. 9).

Taken together, the available structures indicate a possible mechanism for the formation of an amyloid key that is analogous to a previously proposed mechanism for the folding of a Greek key (Fig. 5). The first step of the reaction, two extended segments of a polypeptide chain become paired and form a U-shaped structure. The pairing of the strands may be driven, in an amyloid key, by the association of the aggregation-prone regions, and it may or may not involve the formation of defined β-strands (Fig. 5). This U-shaped structure contains an arc that connects the two extended regions, while the strands of the nascent Greek key are connected by a turn (Fig. 5). In a second step, the paired polypeptide chains fold back onto themselves and establish the final key structure (Fig. 5). We would like to emphasize here that this mechanism does not mean that fibrils with a U-shaped protein fold are able to convert into an amyloid key. Instead, our mechanism means that the polypeptide chain samples, during fibril nucleation, a number of different conformations and that at this stage, and possibly at the level of a monomeric protein, the amyloid key fold becomes formed and stabilized as part of an aggregate. Once established, however, the nucleus structure is able to proliferate its conformation by templated polymerization.

Finally, we noted that our structure and all reported in vitro formed structures from TDP-43 differ from the two fibril morphologies that were purified from patient tissue[40,41] (Supplementary Fig. 9). Hence, TDP-43 can form multiple fibril structures in vivo and there is a spectrum of TDP-43-associated pathologies, which is thought to be associated with these different amyloid fibril structures[27,28]. It is for this reason not possible to exclude that TDP-43 adopts conformations

in vivo that resemble to some of the in vitro formed structures. Nevertheless, the pairing of the two most aggregation-prone regions of the LC domain is so far only seen in the in vitro formed fibrils. In the ex vivo fibrils, the two segments are far apart in the structure and do not form any direct physical interactions (Supplementary Fig. 9). Therefore, the currently available data on TDP-43 fibrils show that in vitro formed fibrils do not necessarily match the structure of ex vivo fibrils in terms of composition and conformation of the fibril protein. In addition, the fold of pathologically relevant fibrils may differ from the one that is adopted most readily by a simple collapse of the most aggregation-prone regions of the sequence but that is more complex and more difficult to establish spontaneously. One reason to establish these specific folds is the higher biological stability of ex vivo fibrils compared with most in vitro formed fibrils[42,43]. That is, ex vivo fibrils may be pathogenic because they resisted the cellular mechanism of clearance and were thus able to accumulate, proliferate, and to become pathogenic in a patient[42,43].

## Methods
### Protein purification and fibril formation
TDP-43 protein was recombinantly expressed as a fusion protein with a TEV cleavage site and an N-terminal hexa-histidine-tag in *Escherichia coli* BL21(DE3) pLysS cells that were transformed with the vector pET24-TDP-43. The protein was purified via nickel affinity chromatography as described previously[33]. In brief, the protein solution after the cell lysis was loaded on a 5 mL HisTrap HP (GE Healthcare) column, which was eluted with 500 mM imidazole in 50 mM Tris buffer, pH 8, containing 0.5 M NaCl and 8 M urea and analyzed with SDS-PAGE (Supplementary Fig. 1a). Pure fractions were dialyzed several times against 40 mM HEPES-KOH, pH 7.4, buffer containing 150 mM KCl, 20 mM MgCl$_2$ and 1 mM dithiothreitol which led to the aggregation of TDP-43. These aggregates were spun down by centrifugation at 13,000 g for 5 min and resuspended in 0.5 mL of 6 M guanidine thiocyanate, 0.5 M NaCl, 50 mM Tris, pH 8, and loaded onto a Superdex 200 10/300 (GE Healthcare) column that was equilibrated in 10 mM HEPES-KOH, pH 7.4, containing 40 mM NaCl. Multimeric species of TDP-43 were eluted in the void volume of the column and monomeric fractions of TDP-43 were eluted at -16 mL (Supplementary Fig. 1b). The elution fractions were analyzed with SDS-PAGE (Supplementary Fig. 1a). The pure monomeric fractions were collected and allowed to self-assemble in the elution buffer for 3 days at room temperature under rotation at

40 rpm on a tube roller (Benchmark Scientific Inc). The resulting fibrils were then used for further analysis.

## TEM analysis of negatively-stained specimens

Formvar and carbon-coated 200 mesh copper EM grids (Electron Microscopy Sciences) were glow discharged for 40 s at 40 mA using a PELCO easiGlow glow discharge cleaning system (Ted Pella). To that end, a 3.5 μL aliquot of the fibril sample was placed on a grid and incubated for 1 min. The sample was blotted with Whatman Grade 595 filter paper (GE Healthcare Life Science). The grid was washed with 10 μL water three times followed by three washing steps with 10 μL 2% (w/v) uranyl acetate. Afterward, the grid was allowed to dry on air for at least 10 min and analyzed using a JEOL1400F TEM. The microscope was operated at an accelerating voltage of 120 kV and images were taken using a CMOS camera (TVIPS TemCam-F216). Two grids per sample were prepared and analyzed.

## TEM analysis of fibrils after platinum side shadowing

To prepare specimens for platinum side shadowing, formvar and carbon-coated 200 mesh copper EM grids (Electron Microscopy Sciences) were glow discharged for 40 s at 40 mA using a PELCO easiGlow glow discharge cleaning system (Ted Pella). A 3.5 μL aliquot of the fibril sample was placed on a grid and incubated for 1 min. Excess sample solution was blotted away with Whatman Grade 595 filter paper (GE Healthcare Life Science), and the grid was washed three times with 10 μL water and allowed to dry for at least 15 min on air. Afterward, the grid was coated with a 1 nm thick layer of platinum, which was evaporated at an angle of 30° using Balzers TKR 010 system. Two grids per sample were prepared and analyzed at 120 kV with a JEOL1400F TEM equipped with CMOS camera (TVIPS TemCam-F216).

## Cryo-EM

C-flat holey carbon grids (CF 1.2/1.3−4C, Electron Microscopy Sciences) were glow discharged using a PELCO easiGlow cleaning system (Ted Pella). A 3.5 μL aliquot of the fibril sample was placed on a grid and incubated for 30 s. The grid was backside blotted for 6 s at a humidity >85% and plunged into liquid ethane using a CP3 plunger (Gatan). A total of ten grids were prepared from the fibril sample and screened with a JEM 2100F (JEOL) TEM that was operated at an accelerating voltage of 200 kV. The high-resolution image data for the cryo-EM reconstruction were recorded with a Titan Krios TEM (Thermo Fisher Scientific) that was operated at an accelerating voltage of 300 kV. Images were taken with a K2 quantum detector (Gatan). The data acquisition parameters are listed in Supplementary Table 1. The fibril width and crossover distances were measured using Fiji[44].

## Helical reconstruction

The recorded multi-frame tiff files were converted to mrcs files and gain corrected using the IMOD software[45]. Using Motioncor2[46], the converted mrcs files were aligned and motion-corrected. CTFFind 4.1[47] was used to estimate the contrast transfer function for the motion-corrected micrograph. Helical 3D reconstruction of the fibrils was performed using Relion 3.1[48].

In case of Morphology-1, 38,441 particles were extracted with a box size of 300 pixels and inter-box distance of 27.64 Å. The reference-free 2D classes were generated with extracted particles using a regularization parameter (T) of 2. An initial 3D reference map was created using 3D initial model tool in Relion 3.1 followed by 3D auto-refinement step with an initial twist of −1.8°, determined by crossover distance, and helical rise of 4.8 Å. A featureless helical structure was created and was used as an initial reference in the next steps. The first 3D classification was performed using the initial 3D reference map with the T value of 4 and three classes. Out of three resulting classes, one class showed Morphology-1a and two classes consisted of Morphology-1b. Particles corresponding to Morphology-1a and Morphology-1b were

selected and re-extracted with a box size of 220 and 230, respectively. For both Morphology-1a and 1b, the re-extracted particles were subjected to multiple rounds of 3D auto-refinement and post-processing. Subsequent Bayesian polishing and following 3D auto-refinement in Relion improved the density map of Morphology-1b, but not of Morphology-1a. To further improvement of the fibril z-axis, atomic models interpreting the reconstructed density maps were transformed into density maps themselves with Chimera and used as reference models for additional 3D auto-refinement with subsequent post-processing and Bayesian polishing.

In the case of Morphology-2, particles were extracted with a box size of 256 pixels and inter-box distance of 27.64 Å, and the reference-free 2D classes were generated with $T = 2$. An initial 3D reference map was generated as described above for Morphology-1 with an initial twist and the helical rise value of −1.4° and 4.8, respectively. The resulting map was used as a reference for the first 3D classification with $T = 4$. The particles from the best class were selected and subjected to 3D auto-refinement followed by 3D classification with one class. The resulting 3D map was used as a reference map for the next rounds of 3D auto-refinement and post-processing. Similar to Morphology-1a and -1b, atomic model interpreting the reconstructed density map were converted into the density map itself and used as a reference model for 3D auto-refinement with following post-processing and Bayesian polishing.

The final 3D maps of the three fibril morphologies were post-processed with a soft-edge mask, and B-factor sharpened using Relion. All reconstruction details are listed in Supplementary Table 1.

## Model building and analysis

The post-processed and masked 3D map of Morphology-2 was used to manually build a model using Coot[49]. A single poly-Ala chain was threaded into the 3D map. The alanine residues were replaced by the suitable residue of the TDP-43 protein. The resulting single polypeptide chain with hydrogen atoms was subjected to several rounds of manual and automated refinements in Coot, during which Ramachandran and beta-sheet restraints were used to maximize the beta-strands content in the structure. Steric clashes between atoms as well as rotamer and Ramachandran outliers were identified with Phenix[50] and corrected using Coot. Once a reasonable density fit was achieved for one polypeptide chain, a fibril stack composed of 18 polypeptide chains was built using the pdbsymm tool implemented in Situs[51]. The described iterative refinement process and validation were repeated for the fibril stack until a satisfactory density to model fit was achieved.

The protein conformation of Morphology-2 was then used as a starting model to interpret the density map of Morphology-1a. A single polypeptide chain was manually fitted to the density map using Coot. The further procedures of structural refinement and model building were similar to those described above for Morphology-2, except that the final fibril stack contained twelve molecules. In the case of Morphology-1b, the protein conformation of Morphology-1a was used as a starting model. As Morphology-1b has an asymmetric cross-section, the two protein stacks of this fibril were modeled individually. Structural refinement and model-building procedures were the same as used in case of Morphology-1a. All modeling details are listed in Supplementary Table 2. The secondary structure assignment was done manually for the fibril models of Morphologies-1a, 1b and 2.

## Prediction of aggregation-prone regions

Seven different computational programs ZipperDB[52], Aggrescan[53], FoldAmyloid[54], Amylpred 2[55], TANGO 2.2[56], PASTA 2.0[57] and Waltz[58] were used to identify the aggregation-prone regions of TDP-43 protein. To perform the prediction, the following settings were used. In case of ZipperDB, if a residue with Rosetta energy below the threshold of −23 kcal/mol was found then the corresponding residue and the following five residues were counted as hits. For Aggrescan, residues

with an a4v value of greater than −0.02 were counted as hits. In case of FoldAmyloid, 5 or more successive residues with a score greater than 21.4 were counted as hits. For AmylPred 2, the consensus method was used to assign a residue as a hit. In TANGO 2.2, an amino acid was counted as a hit that have a score above 5%. For PASTA 2.0, thresholds were kept on the region with 90% specificity which means a residue was counted as a hit if the energy was below the cutoff value of −2.8 PASTA energy units. In the case of WALTZ, residue with a score greater than 0.00 was counted as a hit. An aggregation score of 0 means that none of the programs identifies a given residue as aggregation prone. An aggregation score of 7 means that all seven programs predict a residue as aggregation prone.

## Sample statistics

Wherever appropriate, mean value and errors are presented. Errors represent the standard deviation.

## Reporting summary

Further information on research design is available in the Nature Portfolio Reporting Summary linked to this article.

## Data availability

Cryo-EM datasets were deposited in the Electron Microscopy Public Image Archive under accession codes EMPIAR-11745. The reconstructed cryo-EM maps were deposited in the Electron Microscopy Data Bank with the accession codes: Morphology-1a: EMD-18715; Morphology-1b: EMD-18716; Morphology-2: EMD-18717. The coordinates of the fitted atomic model were deposited in the Protein data bank (PDB) under the accession codes: Morphology-1a: 8QX9; Morphology-1b: 8QXA; Morphology-2: 8QXB. The following previously published coordinates were used in Supplementary Figs. 7 and 8: PDB entries: 6N37, 6N3A, 6N3B, 6N3C, 7Q3U, 7KWZ, 7PY2 and 8CG3. The data that support the findings of this study are available from the corresponding author upon reasonable request. Source Data are provided with this paper.

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

## Acknowledgements
The authors thank Felix Weis (European Molecular Biology Laboratory, Heidelberg), Paul Walther, and Natalie Scheurmann (Ulm University) for technical support. All cryo-EM data were collected at the European Molecular Biology Laboratory, Heidelberg, funded by iNEXT (Horizon 2020, European Union).

## Author contributions
K.S. and F.S. carried out experiments. J.S., M.B., M.B.A.-S., B.H. and A.L. contributed reagents and materials. K.S., A.L., M.S., and M.F. analyzed data. M.F. designed research. K.S. and M.F. wrote the paper. All authors could comment on the manuscript.

## Funding

## Competing interests
The authors declare no competing interests.
