## [Peer review file · Nature Communications]

REVIEWER COMMENTS

Reviewer #1 (Remarks to the Author):

Sharma et al. used cryo-EM to analyze amyloid fibrils of TDP-43 that had been purified recombinantly and aggregated in vitro. They find that the fibrils form a mixture of ultrastructural polymorphs (i.e. fibrils with an essentially shared protofibril fold, but with a different protofibril number and/or protofibril packing interfaces).

Neuronal inclusions containing amyloid fibrils composed of full-length and C-terminal fragments (CTFs) of TDP-43 are the pathological hallmark of amyotrophic lateral sclerosis (ALS), frontotemporal lobar degeneration (FTLD) and limbic predominant age-related TDP-43 encephalopathy (LATE) (Neumann et al. 2006. *Science* 314, 130-133; Nelson et al. 2019. *Brain* 142, 1503-1527).

Previous studies have used cryo-EM to analyze in vitro-generated amyloid fibrils of recombinant CTFs of TDP-43 (Cao et al. 2019. *Nat Struct Mol Biol* 26, 619-627; Li et al. 2021. *Nat Commun* 12, 1620; Kumar et al. 2023. *Nat Neurosci* 26, 983-996). Their structures, like those presented here, are different from the structures of TDP-43 fibrils composed of a mixture of full-length protein and CTFs that are found in the brains of individuals with ALS and FTLD (Arseni et al. 2022 *Nature* 601, 139-143). As such, the relevance of these in vitro-generated fibrils to disease is unclear.

Unlike previous reports of in vitro-generated fibrils, this study used a construct encoding full-length TDP-43. Recombinant full-length TDP-43 has been shown to form fibrils in vitro (Kumar et al. 2023. *Nat Neurosci* 26, 983-996), but structures have not been determined. If the purified TDP-43 was indeed full-length (see major comment 1), this would represent an important advance over previous studies, by showing that fibrils formed of full-length TDP-43 can also be polymorphic.

This work will be of interest to those working on amyloids and interested in their structural and biophysical properties. The figures are well-constructed and the manuscript is generally well-written (but see minor comments). However, major concerns with the quality of the protein preparations and cryo-EM maps (and resulting models) need to be addressed before publication. Please see below for more detailed comments.

Major comments:

1. The quality of the cryo-EM maps should be improved. The PDB validation reports show regions of continuous density running along the helical axis, instead of clearly separated molecular layers, which indicates that the maps may have converged on local minima. Guidance on how to spot and overcome this problem is provided in (Scheres. 2020. *Acta Cryst D* 76, 94-101). In addition, the intermediate resolutions of the maps (~4 Å) may have led to inaccuracies in model building. At present, a relatively modest number of initial particles is used (26,881), suggesting that additional data may improve the maps. The quality of the maps may also be improved by additional rounds of 3D classification, CTF refinement and Bayesian polishing.

2. Are the authors sure that they are dealing with full-length TDP-43, as claimed in the manuscript? Data on the characterisation of the recombinant TDP-43 protein preparations should be shown. At a minimum, this should include chromatograms from the SEC step, as well as SDS-PAGE of the preparations before and after aggregation. This is particularly vital for TDP-43, because it is prone to proteolytic cleavage and aggregation during purification, as well as proteolytic cleavage during in vitro aggregation assays.

3. Please deposit the raw cryo-EM movies to EMPIAR.

Minor comments:

1. The centrifugation step used to pellet fibrils (13,000 g for 5 min) may result in many fibrils being

discarded in the supernatant. For example, TDP-43 fibrils were found in the supernatant following centrifugation at 27,000 g for 10 min in (Arseni et al. 2022 Nature 601, 139-143). The authors should acknowledge this possibility.

2. Line 55, 'Amyloid fibril samples are typically polymorphic and contain multiple fibril morphologies,' does not apply for brain-derived fibril samples, which are often homogeneous. The authors should clarify that this refers to fibril samples generated in vitro.

3. Line 61, 'Examples of different amyloid fibril morphologies in different disease variants,' should also include α -synuclein (Yang et al. 2022. Nature 610, 791-795).

4. Line 83, 'The observed fibril structures are constructed from similarly folded fibril proteins,' and line 121, '...differ in the number of the fibril protein stacks...' are potentially confusing. Do these sentences refer to protofibrils? Please clarify.

5. Line 84, '...exhibiting an amyloid key motif.' This is a central point of the manuscript. Therefore, it would be helpful to define this motif here, or even earlier in the manuscript, rather than waiting until the Results and Discussion sections.

6. Line 140, 'The abundance of the two structural forms varies in the mixed fibrils almost continuously from one to zero.' Does this refer to the proportion of mixed fibrils 0-100%?

7. Line 149, 'The three fibril structures contain essentially the same fibril protein conformation that is formed by 45 residues (Gly304 to Gly348) from the LC domain.' Is there any evidence that the flanking residues to the N- and C-terminus of this region may be structured? It would be helpful for the reader to comment on this.

8. Line 159, 'An unusual structural feature of the fibril core is the absence of buried ion pairs.' This statement needs to be qualified. There are many amyloid fibril structures that lack salt bridges.

9. Line 231, 'Finally, we noted that our structure and all previous in vitro formed structures from TDP-43 differ from a fibril morphology that was purified from a patient.' This study purified TDP-43 fibrils from two patients with ALS and FTL-D-TDP Type B, and two brain regions, prefrontal cortex and motor cortex. This should be corrected.

10. Line 240, 'Therefore, the currently available data on TDP-43 fibrils show that in vitro formed fibrils do not necessarily match the structure of ex vivo fibrils.' It would also be helpful for the reader to point out that the sequence that forms the structured part of the fibril also does not necessarily match (G304-G348 here vs. G281-Q360 in ALS and FTL-D).

11. Line 243 '...but that more complex and more difficult to establish spontaneously.' Should this read, '...but that is more complex and more difficult to establish spontaneously.' ?

12. Line 244 'One reason to establish these specific folds is the higher biological stability of ex vivo fibrils compared with most in vitro formed fibrils.' This claim needs to be supported by data or references.

13. The model of the rotator/shaker used during the aggregation of TDP-43 should be stated to help the reader interpret a rotation of 40 rpm.

14. The program/method used to carry out global refinements of the atomic coordinates, including refining B-factors (i.e. Phenix or REFMAC) should be specified in the Methods.

15. In Figure 1a, it would be helpful to add the position of the NLS

Reviewer #2 (Remarks to the Author):

This short manuscript describes cryo-EM determined three polymorphic structures of amyloid fibrils formed from the recombinant full-length TDP-43. The resolution of these structures (approx. 4 Å) is somewhat below typical resolution of cryo-EM structures previously reported for most other amyloid fibrils. Nevertheless, it is clear that the present structures are fundamentally different (both with regard to the size of the amyloid core and the overall fold) from those previously determined for the recombinant TDP-43 low complexity domain or TDP-43 fibrils extracted from patient brain. Even though high-resolution structures of fibrillar aggregates are always of interest, it is not obvious whether the present structures are of direct physiological/pathological relevance or just represent yet other polymorphs formed in the test tube (see also below).

Specific points:

1. Recombinant full-length TDP-43 is notoriously difficult to purify in a monomeric form. In this context, the authors need to clarify in the Methods section (which is somewhat cryptic in this regard) how the protein (which was applied onto size-exclusion column in the presence of a strong denaturant (6 M thiocyanate quinidine) was eluted from this column, and whether fibrils were formed in the absence or presence of the denaturant. Furthermore, the monomeric state of the purified protein needs to be documented. Those are important issues for the assessment of potential physiological relevance of these structure.

2. The authors should provide a figure that depicts more clearly the density maps. It is impossible to see any relevant details in the small panels of Fig. 2b.

3. SI Figure 2 - The folds of morphology 1-a and morphology 1-b appear to be mirrored images, suggesting that one of them may represent the top view and another one the bottom view? The authors should keep the orientation consistent.

4. SI Figure 2b - There are extra densities near the contact sites of two protofilaments in morphology 1-a and morphology 1-b. These can also be seen in the validation report provided by the authors. The authors may want to comment on these densities.

5. SI Figure 2c - map-model FSC should also be provided. Furthermore, model resolution should be provided in SI table 1 or 2.

6. The authors provide a detailed analysis of beta-sheets formation (line 151-155, figure 4), which likely requires higher resolution model/map than the current results. Moreover, the beta-strand information was actually imposed by the author during the modeling (line 297-299), making it a circular argument.

Minor Issue: Line 272-273 - "An initial 3D reference map was created using 3D initial model in Relion3.1....."

The 3D initial model function in Relion3.1 does not have the function for generating helical models. The authors probably mean the inmodel2d program here?

Revision notes

Reviewer #1 (Remarks to the Author):

Sharma et al. used cryo-EM to analyze amyloid fibrils of TDP-43 that had been purified recombinantly and aggregated in vitro. They find that the fibrils form a mixture of ultrastructural polymorphs (i.e. fibrils with an essentially shared protofibril fold, but with a different protofibril number and/or protofibril packing interfaces).

Neuronal inclusions containing amyloid fibrils composed of full-length and C-terminal fragments (CTFs) of TDP-43 are the pathological hallmark of amyotrophic lateral sclerosis (ALS), frontotemporal lobar degeneration (FTLD) and limbic predominant age-related TDP-43 encephalopathy (LATE) (Neumann et al. 2006. *Science* 314, 130-133; Nelson et al. 2019. *Brain* 142, 1503-1527).

Previous studies have used cryo-EM to analyze in vitro-generated amyloid fibrils of recombinant CTFs of TDP-43 (Cao et al. 2019. *Nat Struct Mol Biol* 26, 619-627; Li et al. 2021. *Nat Commun* 12, 1620; Kumar et al. 2023. *Nat Neurosci* 26, 983-996). Their structures, like those presented here, are different from the structures of TDP-43 fibrils composed of a mixture of full-length protein and CTFs that are found in the brains of individuals with ALS and FTLD (Arseni et al. 2022 *Nature* 601, 139-143). As such, the relevance of these in vitro-generated fibrils to disease is unclear.

Unlike previous reports of in vitro-generated fibrils, this study used a construct encoding full-length TDP-43. Recombinant full-length TDP-43 has been shown to form fibrils in vitro (Kumar et al. 2023. *Nat Neurosci* 26, 983-996), but structures have not been determined. If the purified TDP-43 was indeed full-length (see major comment 1), this would represent an important advance over previous studies, by showing that fibrils formed of full-length TDP-43 can also be polymorphic.

This work will be of interest to those working on amyloids and interested in their structural and biophysical properties. The figures are well-constructed and the manuscript is generally well-written (but see minor comments). However, major concerns with the quality of the protein preparations and cryo-EM maps (and resulting models) need to be addressed before publication. Please see below for more detailed comments.

Major comments:

1. The quality of the cryo-EM maps should be improved. The PDB validation reports show regions of continuous density running along the helical axis, instead of clearly separated molecular layers, which indicates that the maps may have converged on local minima. Guidance on how to spot and overcome this problem is provided in (Scheres. 2020. *Acta Cryst D* 76, 94-101). In addition, the intermediate resolutions of the maps (~4 Å) may have led to inaccuracies in model building. At present, a relatively modest number of initial particles is used (26,881), suggesting that additional data may improve the maps. The quality of the maps may also be improved by additional rounds of 3D classification, CTF refinement and Bayesian polishing.

Response:

We are grateful to this referee for thoroughly analyzing our data and for providing very helpful comments. While the collection of new data was not feasible for us within a reasonable time frame, we managed to further improved the resolution of all the density maps by additional rounds of 3D-auto refine, post processing and Bayesian polishing steps (see method section). The new resolutions range from 3.76 Å to 4.05 Å. More

importantly, they now enable us to see the separation between the molecular layers, which also improved our model building. We have updated related figures accordingly.

2. Are the authors sure that they are dealing with full-length TDP-43, as claimed in the manuscript? Data on the characterisation of the recombinant TDP-43 protein preparations should be shown. At a minimum, this should include chromatograms from the SEC step, as well as SDS-PAGE of the preparations before and after aggregation. This is particularly vital for TDP-43, because it is prone to proteolytic cleavage and aggregation during purification, as well as proteolytic cleavage during in vitro aggregation assays.

Response:

We thank the reviewer for the suggestion. We have taken the feedback into consideration and made the necessary updates to the manuscript. We have now added gel stripes from different steps of the fibril preparation process, including before aggregation (= SEC elution) and after fibril formation, showing a protein size of approximately 43 kDa (see new SI fig.1a). In addition, we included the chromatogram from size exclusion chromatography column (see new SI Fig. 1b). Full-length TDP-43 elutes at approximately 15 ml on a Superdex 200 10/300 column (GE Healthcare). An elution maximum of 15 ml does not correspond to the expected molecular mass of TDP-43. Presumably this is due to interactions of the protein with the column.

We agree to the reviewer that full-length TDP-43 might undergo partial proteolytic cleavage during post-aggregation stages, leaving only a fraction of the C-terminal domain. Further studies would be necessary to investigate in greater detail the proteolytic aging of in vitro fibrils in various experimental storage conditions (temperature, nature of the buffer, pH) and how this process might impact the final fibril structural architecture. However, we believe that such work is clearly outside the scope of the current manuscript, which mainly deals with the structural assessment of the polymorphism of TDP-43 fibrils. To account for the reviewer's comment, we have added a sentence mentioning this possibility in the result section: "although it is possible that proteolytic truncation may occur" after fibril formation.

3. Please deposit the raw cryo-EM movies to EMPIAR.

Response:

Thank you this is done now.

Minor comments:

1. The centrifugation step used to pellet fibrils (13,000 g for 5 min) may result in many fibrils being discarded in the supernatant. For example, TDP-43 fibrils were found in the supernatant following centrifugation at 27,000 g for 10 min in (Arseni et al. 2022 Nature 601, 139-143). The authors should acknowledge this possibility.

Response:

We thank the reviewer for the comment, although we are not sure whether it might have arisen from a misunderstanding of our methodology. The centrifugation step that is mentioned by the referee was performed to collect insoluble TDP-43 protein at an intermediate step of our purification process (between Histrap and size exclusion chromatography). We do not know whether the insoluble TDP-43 collected by centrifugation shows amyloid structure. The precipitated protein was denatured in guanidine thiocyanate, purified as monomeric protein by size exclusion chromatography and allowed to fibrillate in the size exclusion chromatography buffer. This sample was then used for further analysis and it was used without centrifugation.

2. Line 55, 'Amyloid fibril samples are typically polymorphic and contain multiple fibril morphologies,' does not apply for brain-derived fibril samples, which are often homogeneous. The authors should clarify that this refers to fibril samples generated in vitro.

Response:

Thank you for this comment, but many brain-derived fibrils are polymorphic: There are two tau fibril morphologies in Alzheimer patients (Fitzpatrick, et al. 2017, doi: 10.1038/nature23002), two in chronic traumatic encephalopathy (Falcon, et al. 2019, doi: 10.1038/s41586-019-1026-5), two in argyrophilic grain disease (Shi, et al. 2021, doi: 10.1038/s41586-021-03911-7) and two in Pick's disease (Falcon, et al. 2018, doi: 10.1038/s41586-018-0454-y). Alzheimer patients show two morphologies of Aβ42 fibrils (Yang, et al. 2022, doi: 10.1126/science.abm7285-y) and there is even greater heterogeneity in Aβ40 fibrils (Kollmer, et al. 2019, doi: 10.1038/s41467-019-12683-8). α-Synuclein shows two fibril morphologies in multisystem atrophy (Schweighauser, et al. 2020, doi: 10.1038/s41586-020-2317-6) and two fibril morphologies if Lewy bodies are involved, such as in Parkinson's disease (Yang, et al. 2022, doi: 10.1038/s41586-022-05319-3).

3. Line 61, 'Examples of different amyloid fibril morphologies in different disease variants,' should also include α-synuclein (Yang et al. 2022. Nature 610, 791-795).

Response:

Thank you. We now included the α-synuclein paper in the revised version.

4. Line 83, 'The observed fibril structures are constructed from similarly folded fibril proteins,' and line 121, '...differ in the number of the fibril protein stacks...' are potentially confusing. Do these sentences refer to protofibrils? Please clarify.

Response:

Thank you. In the present fibril, a fibril protein stack represents one protofilament. This is expressed in the introduction as "The fibril protein stacks define the fibril protofilaments (PFs), which represent filamentous substructures of amyloid fibrils."

5. Line 84, '...exhibiting an amyloid key motif.' This is a central point of the manuscript. Therefore, it would be helpful to define this motif here, or even earlier in the manuscript, rather than waiting until the Results and Discussion sections.

Response:

Thank you for pointing this out. It very much makes sense to define this motif in the introduction. We now added the following description to our introduction: "In several cases, there was evidence for a fibril protein fold that remotely resembles the Greek key structure in globular proteins. This motif of amyloid fibrils is termed an amyloid key. An amyloid key represents an arrangement of β-strands but in contrast to a Greek key, the strands interact through their side-chains and not through backbone hydrogen bonds."

6. Line 140, 'The abundance of the two structural forms varies in the mixed fibrils almost continuously from one to zero.' Does this refer to the proportion of mixed fibrils 0-100%?

Response:

Thank you. This statement means that mixed fibrils have variable percentages of segments corresponding to the structural forms 1a and 1b. We revised the criticized sentence, which now reads as: "The abundance of the segments corresponding to the

two structural forms (1a and 1b) varies almost continuously within mixed fibrils from one to zero”.

7. Line 149, 'The three fibril structures contain essentially the same fibril protein conformation that is formed by 45 residues (Gly304 to Gly348) from the LC domain.' Is there any evidence that the flanking residues to the N- and C-terminus of this region may be structured? It would be helpful for the reader to comment on this.

Response:

Thank you. While there is evidence for extra density outside the fibril core, this density is very diffuse and we are unable to discern significantly ordered parts in this density, such as flanking residues.

8. Line 159, 'An unusual structural feature of the fibril core is the absence of buried ion pairs.' This statement needs to be qualified. There are many amyloid fibril structures that lack salt bridges.

Response:

We have revised the sentence to now read as: “The fibril core is devoid of buried ion pairs”.

9. Line 231, 'Finally, we noted that our structure and all previous in vitro formed structures from TDP-43 differ from a fibril morphology that was purified from a patient.' This study purified TDP-43 fibrils from two patients with ALS and FTLD-TDP Type B, and two brain regions, prefrontal cortex and motor cortex. This should be corrected.

Response:

Thank you. We have corrected it in the revised manuscript. We now write ‘from patient tissue’ instead of ‘from a patient’.

10. Line 240, 'Therefore, the currently available data on TDP-43 fibrils show that in vitro formed fibrils do not necessarily match the structure of ex vivo fibrils.' It would also be helpful for the reader to point out that the sequence that forms the structured part of the fibril also does not necessarily match (G304-G348 here vs. G281-Q360 in ALS and FTLD).

Response:

Thank you for the suggestion. We have added this information in manuscript. Now, we write “Therefore, the currently available data on TDP-43 fibrils show that in vitro formed fibrils do not necessarily match the structure of ex vivo fibrils in terms of composition and conformation of the fibril protein.”

11. Line 243 '...but that more complex and more difficult to establish spontaneously.' Should this read, '...but that is more complex and more difficult to establish spontaneously.' ?

Response:

Thank you. We have now corrected this in revised version.

12. Line 244 'One reason to establish these specific folds is the higher biological stability of ex vivo fibrils compared with most in vitro formed fibrils.' This claim needs to be supported by data or references.

Response:

Thank you for this comment. We now added references to this statement.

13. The model of the rotator/shaker used during the aggregation of TDP-43 should be stated to help the reader interpret a rotation of 40 rpm.

Response:

The sample was rotated at 40 rpm on a Benchmark TubeRoller (Benchmark Scientific Inc.). This detail has been added to the methods section in the Supplementary Information under the heading ‘Protein purification and fibril formation’.

14. The program/method used to carry out global refinements of the atomic coordinates, including refining B-factors (i.e. Phenix or REFMAC) should be specified in the Methods.

Response:

Thank you. The values of the B-factors are given in SI Table 1 and they were obtained with Relion 3.1.1. We did not sharpen the map using Phenix as it did not result in any significant improvement. This information is now added in the methods section under the heading ‘Helical reconstruction’ and reads as follows: “The final 3D maps of the three fibril morphologies were post-processed with a soft-edge mask, and B-factor sharpened using Relion”.

15. In Figure 1a, it would be helpful to add the position of the NLS.

Response:

Thank you. The position of the NLS is now added in Fig.1a.

Reviewer #2 (Remarks to the Author):

This short manuscript describes cryo-EM determined three polymorphic structures of amyloid fibrils formed from the recombinant full-length TDP-43. The resolution of these structures (approx. 4 Å) is somewhat below typical resolution of cryo-EM structures previously reported for most other amyloid fibrils. Nevertheless, it is clear that the present structures are fundamentally different (both with regard to the size of the amyloid core and the overall fold) from those previously determined for the recombinant TDP-43 low complexity domain or TDP-43 fibrils extracted from patient brain. Even though high-resolution structures of fibrillar aggregates are always of interest, it is not obvious whether the present structures are of direct physiological/pathological relevance or just represent yet other polymorphs formed in the test tube (see also below).

Specific points:

1. Recombinant full-length TDP-43 is notoriously difficult to purify in a monomeric form. In this context, the authors need to clarify in the Methods section (which is somewhat cryptic in this regard) how the protein (which was applied onto size-exclusion column in the presence of a strong denaturant (6 M thiocyanate quinidine) was eluted from this column, and whether fibrils were formed in the absence or presence of the denaturant. Furthermore, the monomeric state of the purified protein needs to be documented. Those are important issues for the assessment of potential physiological relevance of these structures.

Response:

We thank this referee for helpful remarks and suggestions and for alerting us of this problem. We have now revised the methods section to provide more details regarding the purification procedure and hope that this clarifies the uncertainty. In addition, we have included the size exclusion chromatography elution profile and gels from the purification steps (see new SI fig. 1a, 1b). From our size exclusion profile, it is clear that

the protein fraction is devoid of the denaturant as the protein elutes before the denaturant (see the conductivity scale in SI fig. 1b). The protein was aggregated in the size exclusion chromatography buffer, which is 10 mM HEPES-KOH, pH 7.4, containing 40 mM NaCl but no guanidine thiocyanate. We have updated the Methods section to avoid any confusions. Note that the soluble protein was analyzed by solution NMR by us previously (see figure 11 from Shenoy, et al.2019, doi:10.1111/febs.15159), and the solution NMR spectra also revealed a monomeric behavior.

2. The authors should provide a figure that depicts more clearly the density maps. It is impossible to see any relevant details in the small panels of Fig. 2b.

Response:

Thank you for this comment. We now revised Figure 2 and provide a detail map to model comparison in a new Supplementary figure 2.

3. SI Figure 2 - The folds of morphology 1-a and morphology 1-b appear to be mirrored images, suggesting that one of them may represent the top view and another one the bottom view? The authors should keep the orientation consistent.

Response:

Thank you for pointing out this. We are now corrected our mistake.

4. SI Figure 2b - There are extra densities near the contact sites of two protofilaments in morphology 1-a and morphology 1-b. These can also be seen in the validation report provided by the authors. The authors may want to comment on these densities.

Response:

Thank you for this comment. At the current resolution of the 3D maps, we cannot comment on the origin of these extra density. However, it can be possible that they arise due to terminal regions of the fibril protein.

5. SI Figure 2c - map-model FSC should also be provided. Furthermore, model resolution should be provided in SI table 1 or 2.

Response:

Thank you for the suggestion. We now added map-model FSC and model resolution in SI Fig. 3c and SI table 2, respectively.

6. The authors provide a detailed analysis of beta-sheets formation (line 151-155, figure 4), which likely requires higher resolution model/map than the current results. Moreover, the beta-strand information was actually imposed by the author during the modeling (line 297-299), making it a circular argument.

Response:

Thank you for the comment. We now clarify these methodological procedures in the revised methods section under the heading 'Model building and analysis'. We now write 'Ramachandran and beta-sheet restraints were used to maximize the beta-strands content in the structure'. We did not impose specific restraints to defined the beta-sheet in phenix. instead we used beta-sheet restraints to maximize the beta-strand content in our fibril protein.

Minor Issue: Line 272-273 - "An initial 3D reference map was created using 3D initial model in Relion3.1....."

The 3D initial model function in Relion3.1 does not have the function for generating helical models. The authors probably mean the inmodel2d program here?

Response:

Thank you. To create an initial 3D reference map, we used 3D initial model tool in the Relion 3.1 which resulted in a featureless sphere. In the second step, we performed a 3D auto refinement step using this sphere as a reference map and a feature less helical structure was created. This is now clarified in the methods section under the heading 'Helical reconstruction' as follows: "An initial 3D reference map was created using 3D initial model tool in Relion 3.1 followed by 3D auto refinement step with an initial twist of -1.8° , determined by cross-over distance, and helical rise of 4.8 \AA . A featureless helical structure was created and was used as an initial reference in the next steps".

REVIEWERS' COMMENTS

Reviewer #1 (Remarks to the Author):

The authors have addressed most of my comments with the addition of clarifying statements and new data. The SDS-PAGE and chromatogram appears to show that the recombinant TDP-43 is full-length at the time of fibrillization. As such, these results will be of great interest to the field. I have only a few remaining queries:

The quality of the cryo-EM maps has been improved by the author's additional processing. However, the molecular layers now appear somewhat 'flattened,' which may indicate suboptimal helical twist/rise. It may be difficult to improve this without the inclusion of additional data. Nevertheless, the intermediate-resolution of the maps is clearly documented in the manuscript.

Figure 2 shows atomic models, not cryo-EM structures, and so the figure title should be amended accordingly.

It is very difficult to see the cryo-EM maps in SI Figure 2. The authors should use a bolder outline, or a more visible colour. It also appears that the maps have been limited to a zone around the atomic models. This should be made clear in the figure legend.

In SI Figure 3b, densities are visible at the protofibril interfaces of morphologies 1a and 1b. This is an interesting feature not seen in other TDP-43 fibril structures. I recommend including in the manuscript the authors' response to reviewer 2, point 4, that they may arise due to terminal regions of the fibril protein.

Reviewer #2 (Remarks to the Author):

The authors have clarified several technical points, what improves the overall quality of the manuscript. However, a major issue of relatively low resolution of cryo-EM maps still remains largely unresolved. The resolution (now 3.8 – 4.0 Å) is still below the current standards in the field. It appears that improving the resolution would require additional data collection to increase the number of particles used. Authors' response that the collection of new data was not feasible is difficult to understand and accept.

A couple of additional, somewhat less critical issues:

1. Fig. 4a highlights the differences between four different folds in a possibly misleading manner. The major difference seems to be at the level of side chain conformations, and this is derived largely from modeling. If the authors wish to compare side chain conformation, a comparison between maps would be more appropriate at this resolution. Furthermore, for comparing backbone differences, it would be better to use models without side chains.
2. The original maps are still not well depicted in the figures. More clearly presented map figures could potentially clarify some confusions. For instance, Morphology-1a in SI Fig 2 seems to imply that the side chains (and even hydrogens) of G304 and S305 are well resolved in the map. This is highly unlikely at this resolution.

Revision notes

Reviewer #1 (Remarks to the Author):

The authors have addressed most of my comments with the addition of clarifying statements and new data. The SDS-PAGE and chromatogram appears to show that the recombinant TDP-43 is full-length at the time of fibrillization. As such, these results will be of great interest to the field. I have only a few remaining queries:

Response:

We are grateful to this referee for thoroughly analyzing our data and for providing very helpful comments.

The quality of the cryo-EM maps has been improved by the author's additional processing. However, the molecular layers now appear somewhat 'flattened,' which may indicate suboptimal helical twist/rise. It may be difficult to improve this without the inclusion of additional data. Nevertheless, the intermediate-resolution of the maps is clearly documented in the manuscript.

Response:

Thank you. We completely agree with the reviewer's comment.

Figure 2 shows atomic models, not cryo-EM structures, and so the figure title should be amended accordingly.

Response:

Thank you for the comment. We now write 'Cryo-EM 3D maps and molecular models of TDP-43 amyloid fibrils' as panel (a) indeed shows 3D maps.

It is very difficult to see the cryo-EM maps in SI Figure 2. The authors should use a bolder outline, or a more visible colour. It also appears that the maps have been limited to a zone around the atomic models. This should be made clear in the figure legend.

Response:

We thank you for your helpful comment. The SI Figure 2 is now updated with darker cryo-EM maps. We added the information related to the zoning of the maps around the atomic models.

In SI Figure 3b, densities are visible at the protofibril interfaces of morphologies 1a and 1b. This is an interesting feature not seen in other TDP-43 fibril structures. I recommend including in the manuscript the authors' response to reviewer 2, point 4, that they may arise due to terminal regions of the fibril protein.

Response:

Thank you. We now added the information on extra densities visible at the protofibril interfaces of morphologies 1a and 1b in the revised manuscript as follows "In the case of both Morphology-1a and 1b, extra densities were observed near the contact sites of two protofilaments (Supplementary Fig. 3b). At the current resolution of the 3D maps, we cannot comment on the origin of these extra densities. However, it can be possible that they arise due to terminal regions of the fibril protein."

Reviewer #2 (Remarks to the Author):

The authors have clarified several technical points, what improves the overall quality of the manuscript. However, a major issue of relatively low resolution of cry-EM maps still remains largely unresolved. The resolution (now 3.8 – 4.0 Å) is still below the current standards in the field. It appears that improving the resolution would require additional data collection to increase the number of particles used. Authors' response that the collection of new data was not feasible is difficult to understand and accept.

Response:

We thank this referee for the remarks. Preparing new samples, collecting new data, and following the reconstruction of 3D maps will take at least 5-6 months if everything goes well. The main reason hereof is that we do not have an in-house high-resolution cryo-EM facility to collect more data and we depend on the access schemes of external facilities. Moreover, previous data collection was done using INEXT funding which is no longer available.

A couple of additional, somewhat less critical issues:

1. Fig. 4a highlights the differences between four different folds in a possibly misleading manner. The major difference seems to be at the level of side chain conformations, and this is derived largely from modeling. If the authors wish to compare side chain conformation, a comparison between maps would be more appropriate at this resolution. Furthermore, for comparing backbone differences, it would be better to use models without side chains.

Response:

Thank you for this comment. We agree with the referee that the differences in the side chains may have arisen during modelling. This is now expressed in the figure legend as “note that differences in the side chain geometries may have arisen from the modelling”. We further noted that the figure is referred to in the main text misleadingly and the figure reference was changed. The purpose of this figure is to demonstrate the structural similarity and not to pinpoint structural differences.

2. The original maps are still not well depicted in the figures. More clearly presented map figures could potentially clarify some confusions. For instance, Morphology-1a in SI Fig 2 seems to imply that the side chains (and even hydrogens) of G304 and S305 are well resolved in the map. This is highly unlikely at this resolution.

Response:

Thank you for pointing out this. We have updated the SI fig. 2 and the visibility of cryo-EM maps is improved. Moreover, in all figures, we removed the hydrogens in the fibril models to create less confusion because, at the current resolution of 3D maps, hydrogens are not resolved.